# High Energy Storage Density in Nanocomposites of P(VDF-TrFE-CFE) Terpolymer and BaZr_0.2_Ti_0.8_O_3_ Nanoparticles

**DOI:** 10.3390/ma15093151

**Published:** 2022-04-27

**Authors:** Yusra Hambal, Vladimir V. Shvartsman, Ivo Michiels, Qiming Zhang, Doru C. Lupascu

**Affiliations:** 1Institute for Materials Science and Center for Nanointegration Duisburg-Essen (CENIDE), University of Duisburg-Essen, 45141 Essen, Germany; yusra.hambal@uni-due.de (Y.H.); ivo.michiels@uni-due.de (I.M.); doru.lupascu@uni-due.de (D.C.L.); 2School of Electrical Engineering and Computer Science, Materials Research Institute, The Pennsylvania State University, University Park, State College, PA 16802, USA; qxz1@psu.edu

**Keywords:** energy storage, polymer composite, relaxor, P(VDF-TrFE-CFE), barium titanate, nanoparticles

## Abstract

Polymer materials are actively used in dielectric capacitors, in particular for energy storage applications. An enhancement of the stored energy density can be achieved in composites of electroactive polymers and dielectric inorganic fillers with a high dielectric permittivity. In this article, we report on the energy storage characteristics of composites of relaxor terpolymer P(VDF-TrFE-CFE) and BaZr_0.2_Ti_0.8_O_3_ (BZT) nanoparticles. The choice of materials was dictated by their large dielectric permittivity in the vicinity of room temperature. Free-standing composite films, with BZT contents up to 5 vol.%, were prepared by solution casting. The dielectric properties of the composites were investigated over a wide range of frequencies and temperatures. It was shown that the addition of the BZT nanoparticles does not affect the relaxor behavior of the polymer matrix, but significantly increases the dielectric permittivity. The energy storage parameters were estimated from the analysis of the unipolar polarization hysteresis loops. The addition of the BZT filler resulted in the increasing discharge energy density. The best results were achieved for composites with 1.25–2.5 vol.% of BZT. In the range of electric fields to 150 MV/m, the obtained materials demonstrate a superior energy storage density compared to other P(VDF-TFE-CFE) based composites reported in the literature.

## 1. Introduction

Environmentally-friendly renewable energy sources are making an increasing contribution to the production of electrical energy. However, due the intermittent nature of their output, there is a growing demand for effective electrical energy storage technologies. Among the possible types of devices used for electrical energy storage, dielectric capacitors feature very fast (nanoseconds to microseconds) charging/discharging times, which provide high power density and make them useful for applications where the rapid delivery of energy is necessary, e.g., pulsed power systems [1]. Due to their ability to sustain high electric fields, polymers were commercially used in dielectric capacitors for decades [2]. Of particular interest are electroactive polymers, such as ferroelectric polyvinylidene fluoride P(VDF) and its copolymers with trifluoroethylene (TrFE), hexafluoropropylene (HFP), chlorofluoroethylene (CFE), and chlorotrifluoroethylene (CTFE) [3,4,5,6,7]. These materials show large relative dielectric permittivity that allows to achieve significantly higher values of stored electrical energy than in the case of linear dielectrics. The stored electrical energy density, *U_stored_*, is calculated by integrating the area between the charging branch of a dielectric displacement–electric field hysteresis loop and the dielectric displacement axis,
(1)$$U_{stored}=\int_{0}^{D_{max}}E\left(D\right)dD$$
with the dielectric displacement, *D*(*E*), given by the polarization $P\left(E\right)$ of the material and the dielectric permittivity of free space $D\left(E\right)=\varepsilon_{0}\cdot E+P\left(E\right)$. Here for simplicity, we consider the case of an isotropic material, which roughly corresponds to our experimental condition of a polycrystalline polymer, and the polarization measured collinear to the applied electric field. Usually, in ferroelectrics, $P\left(E\right)\gg \;\varepsilon_{0}\cdot E$, and $D\left(E\right)\approx P\left(E\right)$. Therefore, the stored energy density is estimated as
(2)$$U_{stored}=\int_{0}^{P_{max}}E\left(P\right)dP$$

For practical applications, the part of the stored energy that is reused is important. It is called recoverable or discharge energy. The corresponding energy density, *U_discharge_*, is defined by the area between the discharging part of the *P–E* hysteresis loop and the polarization axis.
(3)$$U_{discharge}=\int_{P_{rem}}^{P_{max}}E\left(P\right)dP$$

Due to losses related to the leakage current and polarization switching, *U_discharge_* < *U_stored_*. The difference between *U_discharge_* and *U_stored_* is given by the polarization hysteresis loop area. The energy storage efficiency, *η*, is defined as the ratio between the discharge and stored energies of the capacitor.
(4)$$\eta =U_{discharge}/U_{stored}$$

In the case of ferroelectric materials, in spite of the large, incorporated energy density, the efficiency, *η*, is usually small, due to both the large remanent polarization and the large coercive electric field. Therefore, materials with a high polarizability (large *P_max_*), together with small remanent polarization, *P_rem_*, and slim hysteresis loops are more suitable for application. Such a combination of properties is found in relaxor ferroelectrics or, in short, relaxors [8,9,10].

Relaxors have a very particular polar structure. Namely, the transition into the long-range ordered ferroelectric state is suppressed by the structural or charge disorder inherent to these materials [11]. However, a short-range polar order within polar nanoregions (PNRs) arises at high temperatures [12]. The PNRs of sizes of a few nanometers can be described as superparaelectric dipoles [13]. At high temperatures they are dynamic and are easily ordered by an electric field, facilitating a large polarizability of relaxors. When the field is removed, the PNRs return to a disordered state, resulting in a small remanent polarization.

In the case of ferroelectric polymer materials, such as P(VDF-TrFE), the relaxor behavior can be achieved by the incorporation of a bulky monomer, such as hexafluoropropylene (HFP), chlorotrifluoroethylene (CTFE), or chlorofluoroethylene (CFE), to co-polymer chains [14,15,16,17]. In particular, terpolymers P(VDF-TrFE-CFE), with the VDF/TrFE molar ratio below 75/25 and the molar amount of CFE > 4 mol.%, exhibit relaxor behavior with a broad and frequency-dependent peak in the temperature dependence of the dielectric permittivity [14,15,18].

Relatively large room temperature dielectric permittivity, small remanent polarization, and slim hysteresis loops make relaxor P(VDF*_x_*-TrFE_1−*x*_-CFE*_y_*) terpolymers attractive for energy storage applications [3,18,19,20,21]. Nevertheless, the dielectric permittivity of the polymers fall short of the ones of inorganic relaxors. One of the popular strategies to combine the advantages of both material classes is to make composites consisting of a polymer matrix with ceramic nanofillers [20,21]. Such composites combine the flexibility, light weight, and high dielectric strength of polymers with the large dielectric permittivity, and piezoelectric and pyroelectric coefficients of inorganic ferroelectrics. It is shown that the introduction of ferroelectric nanoparticles, such as BaTiO_3_, Pb(Zr,Ti)O_3_, or TGS, into polymers yields capacitors with significantly enhanced piezoelectric and pyroelectric coefficients, as well as stored energy density [20,21,22,23,24,25,26,27,28,29,30,31,32]. The morphology of the filler, its volume fraction, the surface properties of the filler, the properties of the polymer–filler interface, and the distribution of the filler in the polymer matrix play important roles in optimizing the energy storage properties. Last year, a machine learning approach has been adopted to search for the optimized filler characteristics [33].

Recently, we have reported on the dielectric, ferroelectric, and energy storage properties of several new compositions of P(VDF-TrFE-CFE) terpolymers [18]. In this work, we investigated the energy storage properties of composites based on P(VDF-TrFE-CFE) 64.8/35.2/7.8 and 68/32/8.5 terpolymers and BaZr_0.2_Ti_0.8_O_3_ (BZT) nanoparticles. The substitution of 20% of Ti^4+^ with Zr^4+^ shifts the maximum of the dielectric permittivity from approximately 120 °C in pure BaTiO_3_ to room temperature [34]. Moreover, it induces the relaxor behaviour, allowing large values of the dielectric permittivity in a broad range around room temperature [35].

## 2. Materials and Methods

Terpolymer P(VDF*_x_*-TrFE_1−*x*_-CFE*_y_*) 64.8/35.2/7.8 and 68/32/8.5 powders were purchased from Piezotech (Pierre-Bénite, France). Zirconium-doped barium titanate, BaZr_0.2_Ti_0.8_O_3_ (BZT), nanoparticles were used as nanofillers for the polymer nanocomposites. The concentration of zirconium was set to 20 at.%, which is optimal to match the maxima of the dielectric permittivity of the P(VDF-TrFE-CFE) matrix and the BZT filler. The nanoparticles were synthesized in an autoclave via the hydrothermal route [36,37]. Details of the synthesis of nanoparticles and their characterization are published elsewhere [38]. The mean size of BZT nanoparticles was 150 nm.

Free-standing pure polymer, and nanocomposite films with varying amounts of BZT nanofiller (1.25 vol.% to 5 vol.%), were prepared using the solution casting method. The targeted amount of nanoparticles was added to the polymer solution and the suspension was stirred for 2 h. The polymer/nanoparticle suspension was then drop coated on to a glass substrate (Corning Inc., Corning, NY, USA). The coating was dried at 60 °C for 20 h, then annealed under vacuum (260 mbar) at 100 °C for 8 h, and slowly cooled down to room temperature. The film with the substrate was then immersed in distilled water, peeled off, and dried using a fibreless tissue. The final thickness of the freestanding films was around 20 μm. More details on the synthesis and characterization of BZT nanoparticles and free-standing nanocomposite films are found elsewhere [38].

For electrical characterization, silver electrodes were sputtered onto both sides of the films using a Cressington sputter coater 208 HR. The approximate thickness of the silver layer was 50 nm. The dielectric permittivity was measured in a frequency range of 10^3^–10^6^ Hz, using a Solartron 1260 impedance analyzer with a dielectric interface 1296. The measurements were performed upon heating, as well as upon cooling, within a temperature range from 270 K to 370 K. The polarization hysteresis loops were measured using a TF Analyzer 2000 (Aixacct, Aachen, Germany). A triangular wave function with the frequency of 10 Hz was employed for the application of both unipolar and for bipolar electric fields. The energy storage properties were evaluated by analyzing the polarization hysteresis loops measured under unipolar field with varying amplitudes.

## 3. Results

### 3.1. Dielectric Properties

Figure 1 shows the temperature dependences of the real part of dielectric permittivity and dielectric loss tangent of the neat polymer P(VDF-TrFE-CFE) 68/32/8.5 sample, and the nanocomposites with the different filler contents, measured upon cooling from 370 K to 270 K. The neat polymer, as well as its nanocomposites, show a broad peak of the permittivity curves around room temperature. The positions of the permittivity peaks shift towards higher temperatures with frequency, the values of the dielectric permittivity decrease, while the dielectric loss tangent changes incrementally. The broad dielectric peaks, and their frequency dispersion, validate the relaxor behavior of P(VDF-TrFE-CFE) 68/32/8.5 and its nanocomposite films.

In Figure 2a,b, the real part of dielectric permittivity and dielectric loss tangent of the neat polymer sample and its nanocomposites are compared for the frequency 1 kHz. The neat polymer exhibits a maximum dielectric permittivity of about 55, which progressively increases upon the addition of the BZT nanofillers. The dielectric permittivity of the nanocomposite film with 5 vol.% of BZT nanofiller is almost twice that of the neat polymer, reaching approximately 100 at the maximum. This trend is in agreement with previous reports [38,39]. The increase in dielectric permittivity signifies that the saturation polarization is achieved faster in nanocomposites than in the neat polymer [24,40]. The permittivity values of nanocomposite films at 1 kHz show a slight increase at higher temperature. The slight increment of the dielectric permittivity of P(VDF-TrFE-CFE) nanocomposites at low frequencies (≤1 kHz), and at higher measuring temperature, has been previously observed as well [41]. This increment can be attributed to the presence of different charges on the surface of BZT nanoparticles, and at the polymer/nanoparticle interface, which becomes significant with rising temperature.

At the same time, the relaxor characteristics of the terpolymer seem to be less affected by adding the nanofillers. The strong frequency dispersion of the dielectric permittivity is characteristic for relaxors. It is a manifestation of the slowing down of PNRs dynamics upon cooling, which results in a transition to a short-range ordered polar glass state at the so-called freezing temperature, *T_f_*. Figure 3 shows the freezing temperature, *T_f_*, for the films with different contents of the BZT filler. The freezing temperature was estimated from the temperature dependences of the real part of dielectric permittivity, according to the Vogel–Fulcher relation [13],
(5)$$f=f_{0}exp\left(-\frac{E_{a}}{k_{B}\left(T_{m}\left(f\right)-T_{f}\right)}\right)$$
where *T_m_*(*f*) is the temperature of the maximum of the dielectric permittivity at the frequency *f*, *E_a_* is the activation energy, and *k_B_* is the Boltzmann constant. One can see that the freezing temperature did not change substantially upon increasing nanofiller content, except for the sample with 1.25 vol.% of BZT.

A similar trend is observed for composite films based on the P(VDF-TrFE-CFE) 64.8/35.2/7.8 (Figure 2c,d) terpolymer. The temperature dependences of the dielectric permittivity for these composites, measured at different frequencies, are shown in the Appendix A). The relative increase of the dielectric permittivity at the maximal content of the filler is 70% for P(VDF-TrFE-CFE) 64.8/35.2/7.8.

Appendix A demonstrate the frequency dependences of *ac* electrical conductivity measured at different temperatures. Both composite systems exhibit similar behavior. At temperatures between 320 and 370 K, the ac conductivity increases linearly with frequency, and the temperature variation of the conductivity is relatively weak. Below 310 K, a deviation from the linear dependence becomes visible, and the conductivity drops noticeably. Similar behavior is reported for other polymers [42]. We can attribute this to the slowing down of the PNRs relaxation, which occurs in the same temperature range.

We observed that in the composite films, the conductivity increases with the increasing amount of filler (Appendix A). This correlates with the trend reported in the literature [43,44]. In addition, the P(VDF*_x_*-TrFE_1−*x*_-CFE*_y_*) 64.8/35.2/7.8 terpolymer has a larger ac conductivity than the P(VDF*_x_*-TrFE_1−*x*_-CFE*_y_*) 68/32/8.5 terpolymer.

### 3.2. Polarization Hysteresis Loops

Figure 2c,f shows the bipolar polarization hysteresis loops, measured for the neat polymer and the composite films, at room temperature. The maximal field applied to the composite samples is 75 MV/m. At higher fields, a dielectric breakdown occurred. The neat P(VDF-TrFE-CFE) 68/32/8.5 terpolymer shows a paraelectric-like loop with a negligible remanent polarization, and small maximal polarization (1.25 µC/cm^2^). However, adding 1.25 vol.% of BZT results in a slim hysteresis loop, typical for relaxors with a large maximal polarization (*P_max_* = 6.3 μC/cm^2^), but small remanent polarization. Further increasing the BZT content leads to a decrease in the maximal polarization, to 5 μC/cm^2^. Concurrently, the remanent polarization and the coercive electric field increase, manifesting that the ferroelectric state becomes more stable. For the neat terpolymers, a high electric field induces the transition into a ferroelectric state, which is manifested by the appearance of double hysteresis loops [18]. For the P(VDF-TrFE-CFE) 68/32/8.5 polymer, such a transition occurs at a field of more than 90 MV/m. The insert in Figure 2c shows the hysteresis loop for the neat polymer taken at a field, with an amplitude of 150 MV/m. In this case, the maximal polarization (*P_max_* = 4 µC/cm^2^) is still less than in the composite films at the much lower electric field of 75 MV/m.

Similarly, for the composites based on the P(VDF-TrFE-CFE) 64.8/35.2/7.8 terpolymer, increasing the BZT content results in increasing the maximal polarization (Figure 2f). At the same time, the remanent polarization did not change significantly. The composition with 5 vol% BZT shows a larger *P_max,_* together with a smaller area of the hysteresis loops, as compared to the composite based on the P(VDF-TrFE-CFE) 68/32/8.5 terpolymer.

Thus, introducing the BZT nanoparticles increases the polarizability of the terpolymer matrix. This observation correlates with the dielectric data where the dielectric permittivity of the nanocomposites is significantly higher than that of the terpolymers.

### 3.3. Energy Storage Properties

To evaluate the energy storage characteristics of the sample, unipolar polarization hysteresis loops were measured (Figure 4). The stored and discharge electrical energy densities were calculated from the charge and discharge branches of the hysteresis loops according to Equations (2) and (3), respectively. The energy storage efficiency, *η*, was calculated from these data using Equation (4). The dielectric strength of the composite films under the unipolar load is higher than under a bipolar load. The composites based on the P(VDF-TrFE-CFE) 64.8/35.2/7.8 terpolymer could withstand relatively large electric field, for some samples even up to 150 MV/m. However, for the composites based on the P(VDF-TrFE-CFE) 64.8/35.2/8.5 terpolymer, dielectric breakdown occurred at a field below 100 MV/m.

In Figure 5, the discharge energy density, as well as the charge–discharge energy efficiency, of the nanocomposites of P(VDF-TrFE-CFE) 64.8/35.2/7.8 and 68/32/8.5 at room temperature are presented as a function of maximum applied electric field. For both P(VDF-TrFE-CFE) compositions, the discharge energy density increases upon the addition of BZT nanoparticles. At 75 MV/m, the discharge energy density of the P(VDF-TrFE-CFE) 64.8/35.2/7.8 nanocomposites with the addition of 2.5 vol.% and 5 vol.% BZT nanoparticles is approximately 2.5 times more than that of the pure terpolymer. However, at the higher electric fields, the best energy storage is achieved by the composite with 1.25 vol.% BZT, reaching the value of 4.5 J/cm^3^ at a field of 125 kV/cm. The relative decrease in the discharge energy density in the composites with the larger nanofiller content reflects the situation when the saturation polarization is reached, but the remanent polarization and the area of the hysteresis loop continue to increase (Figure 2f). Correspondingly, the storage efficiency drops to 36% (at 125 kV/cm) for the composite with 5 vol.% of the nanofiller, while for the sample with 1.25 vol.% of nanofiller it is about 53% at the same field (Figure 5b).

A similar trend is observed for the P(VDF-TrFE-CFE) 68/32/8.5 nanocomposites. In these materials, the largest discharge energy density of ~2.8 J/cm^3^ (at 75 kV/cm) is observed in the composite with 2.5 vol.% of BZT, which is approximately two times larger than in the neat terpolymer. The composite with 5 vol.% of BZT shows a lower energy density of ~2.5 J/cm^3^, together with a significantly smaller storage efficiency of 53%.

## 4. Discussion

We have observed that the incorporation of nanoparticles with a larger dielectric permittivity increases the total dielectric permittivity of the composites. There are several models to calculate the dielectric permittivity of multicomponent materials. For a composite consisting of spherical particles, with permittivity $\varepsilon_{1}$ homogeneously distributed in a matrix with permittivity $\varepsilon_{2}$, the effective permittivity $\varepsilon_{eff}$ can be calculated, for example, using the Lichtenecker model (Equation (6)) [45] or the Maxwell–Garnett model (Equation (7)) [46]:(6)$$ln\varepsilon_{eff}=xln\varepsilon_{1}+\left(1-x\right)ln\varepsilon_{2}$$
(7)$$\varepsilon_{eff}=\varepsilon_{2}\frac{2x\left(\varepsilon_{1}-\varepsilon_{2}\right)+\left(\varepsilon_{1}+2\varepsilon_{2}\right)}{\left(\varepsilon_{1}+2\varepsilon_{2}\right)-x\left(\varepsilon_{1}-\varepsilon_{2}\right)}$$
where *x* is the volume fraction occupied by the particles. Figure 4c,d shows the experimental values of the dielectric permittivity of the studied composites, measured at a temperature of 300 K and a frequency of 1 kHz, together with the dependences calculated according to Equations (6) and (7). The dielectric permittivity of BZT particles was taken from the data published in Ref. [35]. It is seen that the measured dielectric permittivity goes beyond the predictions of simple mixture models. This means interface effects may play a role. It was reported that in dipolar polymer composites, dipoles in interfacial regions experience reduced constraints in response to an applied electric field, leading to a significantly enhanced dielectric response [47,48]. It was suggested that filler particles generate local and nanostructure changes that weaken hydrogen bonds and increase distances between polymer chains, reducing the local constraints on the ordering of polymer dipoles along the applied field [48].

For both P(VDF-TrFE-CFE) 64.8/35.2/7.8 and 68/32/8.5 polymers, the incorporation of BZT nanoparticles leads to the increased polarizability of the composites, which is manifested in larger values of the dielectric permittivity and maximal polarization, under both bipolar and unipolar electric field cycling. Naturally, these factors improve the discharge energy density, which first increases with an increasing amount of BZT nanoparticles. The nanocomposites with 2.5 vol.% of BZT based on P(VDF-TrFE-CFE) 64.8/35.2/7.8 and 68/32/8.5 terpolymers show a comparable discharge energy density of 2.6–2.8 J/cm^3^ at the same field of 75 kV/cm. The corresponding storage efficiency lies in a range of 65–70%. The charge–discharge efficiency of the 64.8/35.2/7.8 nanocomposites is higher than that of the 68/32/8.5 nanocomposites, since for the neat 68/32/8.5 polymer double *P–E* loops are typical [18], which additionally increases the hysteresis area.

A particular feature of P(VDF-TrFE-CFE) terpolymers is a field induced phase transition from a relaxor to a ferroelectric state [18]. This results in an increased remanent polarization (when the field is removed, only a part of the ferroelectric region transforms back to the relaxor state), and a widening of the hysteresis loop area when the amplitude of the applied field increases. Correspondingly, the relation between the discharged and stored energy, the storage efficiency, also decreases (Figure 5b,d).

We also observed that in the composites with 5 vol.% of the BZT filler, both the discharge energy density and storage efficiency are lower than in other composites, especially at a higher electric field. This again correlates with the larger hysteresis for these composites, which can be explained in terms of surface charges accumulated at the BZT/polymer interfaces. These charges contribute to the dielectric losses and leakage current, which become remarkable with higher temperature, even at a low electric field, as seen in dielectric permittivity curves (Figure 1 and Figure 2), or at higher electric fields in the polarization hysteresis curves. The increased volume of the nanofiller increases the interface area and, correspondingly, the amount of these surface charges.

In Table 1, the energy storage parameters of the composites under study are compared with the results for other composites, based on P(VDF-TrFE-CFE) terpolymers reported in the literature. Most of the research focuses on materials containing fillers in the form of nanofiber or nanosheets, as these composites are expected to have higher dielectric strength [26,27,28,29,30,31,32]. Nevertheless, one can see that, at moderate electric fields (100–150 kV/cm), the composites with nanoparticles outperform the composites with nanowires or nanosheets, demonstrating a superior energy storage capacity at a relatively low volume fraction.

## 5. Conclusions

Polymer–inorganic free-standing composite films of P(VDF-TrFE-CFE) 64.8/35.2/7.8 and 68/32/8.5 terpolymers and BaZr_0.2_Ti_0.8_O_3_ nanoparticles show significantly enhanced discharge energy density, in comparison to the neat polymer films. This enhancement is directly related to the increased polarizability of the composite sample, which is manifested in their large dielectric permittivity. Moreover, in the studied electric field range (below 125 MV/m), the discharge energy density for the obtained composites significantly exceeds the values for composites based on P(VDF-TrFE-CFE) reported in the literature. This demonstrates that the polymer–nanoparticle composites may be more promising for energy storage applications than composites with other filler morphology (nanowires, nanosheets), at least in a range of a moderate electric field.

## Figures and Tables

**Figure 1 materials-15-03151-f001:**
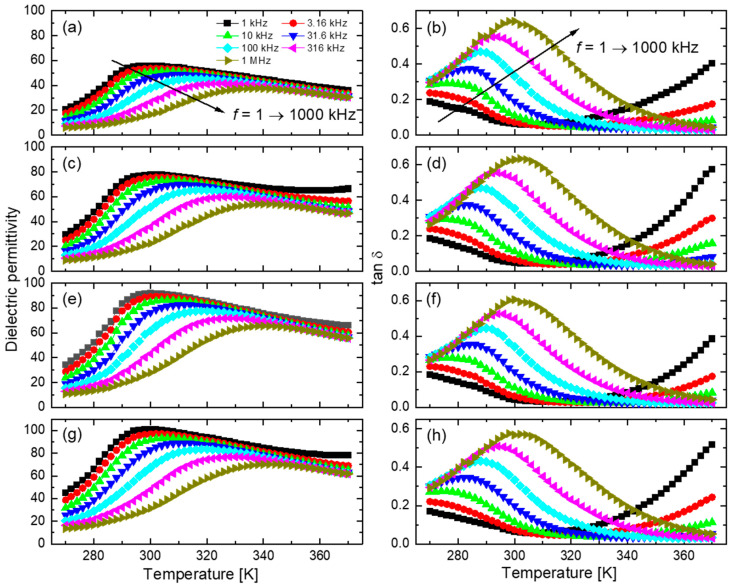
Temperature dependences of the real part of dielectric permittivity and dielectric loss tangent of P(VDF-TrFE-CFE) 68/32/8.5 with varying BZT nanoparticle contents, (**a**,**b**) 0 vol.%, (**c**,**d**) 1.25 vol.%, (**e**,**f**) 2.5 vol.%, and (**g**,**h**) 5 vol.%.

**Figure 2 materials-15-03151-f002:**
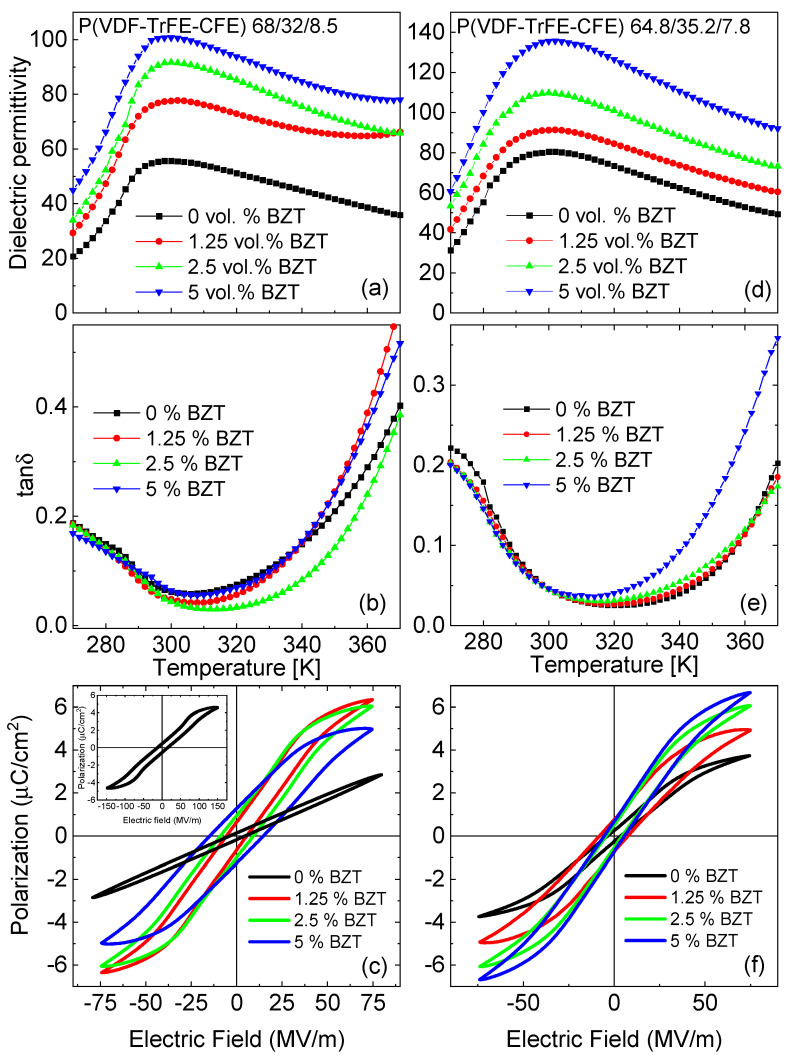
Temperature dependences of the real part of dielectric permittivity (**a**,**d**), and dielectric loss tangent (**b**,**e**) measured at a frequency of 1 kHz, and bipolar polarization hysteresis loops at room temperature (**c**–**f**) for composites based on P(VDF-TrFE-CFE) 68/32/8.5 (**a**–**c**), and P(VDF-TrFE-CFE) 64.8/35.2/7.8 (**d**–**f**) terpolymers.

**Figure 3 materials-15-03151-f003:**
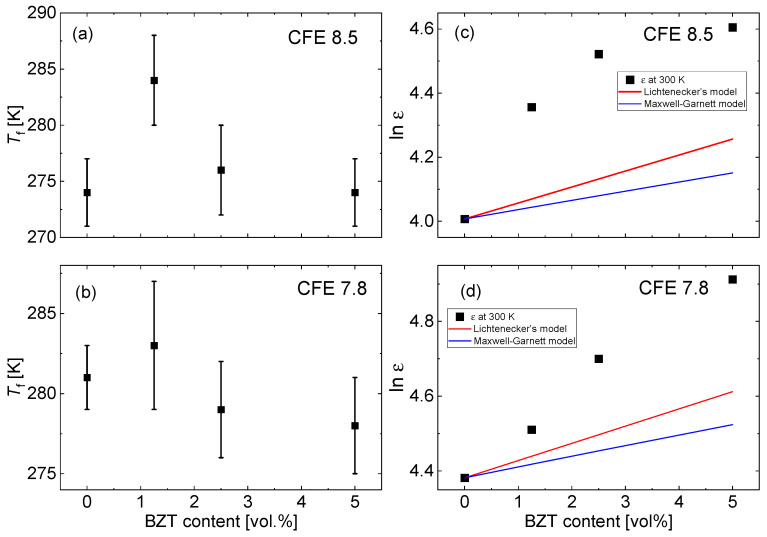
The freezing temperature, *T_f_*, according to the Vogel–Fulcher fit of *T_m_*(*f*) dependences for P(VDF-TrFE-CFE) 68/32/8.5 (**a**), and P(VDF-TrFE-CFE) 64.8/35.2/7.8 (**b**) based composites with different BZT contents. Dependences of the dielectric permittivity (at T = 300 K and f = 1 kHz) on the BZT content for P(VDF-TrFE-CFE) 68/32/8.5 (**c**), and P(VDF-TrFE-CFE) 64.8/35.2/7.8 (**d**) based composites. The straight lines show the dielectric permittivity of the composites calculated according to the Lichtenecker and Maxwell–Garnett models.

**Figure 4 materials-15-03151-f004:**
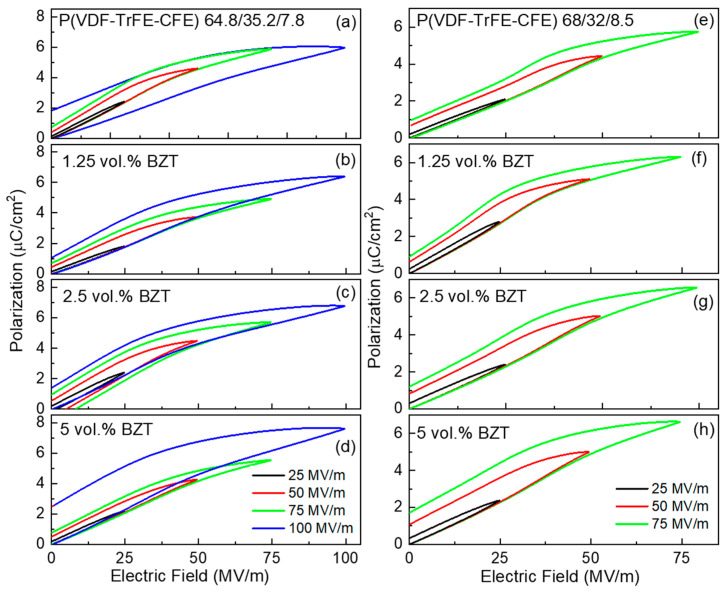
Unipolar polarization hysteresis loops of composite films based on P(VDF-TrFE-CFE) 64.8/35.2/7.8 (**a**–**d**), and 68/32/8.5 (**e**–**h**) terpolymers measured under triangular field of 10 Hz at room temperature.

**Figure 5 materials-15-03151-f005:**
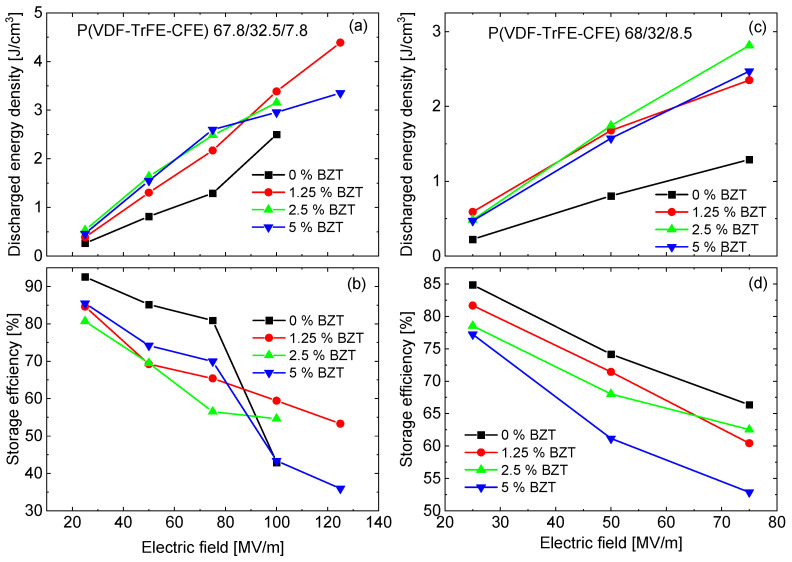
Discharge energy density (**a,c**) and storage efficiency (**b**,**d**) for the composite films based on P(VDF-TrFE-CFE) 64.8/35.2/7.8 (**a**,**b**), and 68/32/8.5 (**c**,**d**) terpolymers.

**Table 1 materials-15-03151-t001:** Comparison of discharge energy density in composites based on relaxor P(VDF-TrFE-CFE) terpolymers.

Filler	Morphology	Fraction	Electric Field (MV/m)	Discharged Energy Density (J/cm^3^)	Reference
Ba(Zr,Ti)O_3_	Nanoparticles	2.5 vol.%	75	2.8	This work
Ba(Zr,Ti)O_3_	Nanoparticles	1.25 vol.%	125	4.5	This work
Ba(Zr,Ti)O_3_	Intralayer		75/125	1.3/2.2	[31]
SrTiO_3_	Nanofibers	7 vol.%	75/125	1.0/2.2	[30]
BaTiO_3_	Nanowires	17.5 vol.%	75	1.6	[26]
BaTiO_3_	Nanowires	17.5 vol.%	100	2.0	[27]
BaTiO_3_@TiO_2_	Nanofibers	10 wt.%	150	2.9	[28]
(Ba,Ca)(Zr,Ti)O_3_	Nanofibers	3 vol.%	75/125	1.0/1.3	[29]
Graphene	Sheets	0.5 wt.%	125	1.2	[32]

## Data Availability

The data presented in this study are available on request from the corresponding author.

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
