# Peer review of "High Energy Storage Density in Nanocomposites of P(VDF-TrFE-CFE) Terpolymer and BaZr0.2Ti0.8O3 Nanoparticles"

_materials, 2022, doi:10.3390/ma15093151_

Round 1

Reviewer 1 Report

The subject of the present work is interesting scientifically and also has technological importance. The manuscript is well written and organized.

Minor points to be considered by the authors:

1). Authors use the term "dielectric permittivity" to describe the "real part of dielectric permittivity".

2). Please upon the physical origin of the relaxation process shown in Fig. 1.

3). Which is the glass transition temperature of the examined systems, and what is the influence of filler on it?

4). How temperature affects the storage efficiency?

5). Since Eq. (5) and (6) fail by far to describe the obtained data (Fig. 3), which is the reason of including them in the paper.

Concluding the present work can be accepted for publication after minor revision.

Author Response

  1. Authors use the term "dielectric permittivity" to describe the "real part of dielectric permittivity".

Thank you. It was corrected.

  1. Please upon the physical origin of the relaxation process shown in Fig. 1.

 The relaxation of the dielectric permittivity visible on the dependences shown in Fig.1 is characteristic for the relaxor state of terpolymers P(VDF-TrFE-CFE). According to the literature it is related to dynamic slowing down of polar nanometer size regions (PNRs). These PNRs with the size of a few nm form in relaxors at temperatures above the temperature of maximum of the dielectric permittivity. On cooling down the polarization remains short range ordered within PNRs, and a transition to a long-range ordered ferroelectric state does not occur. Instead, slowing down or freezing of the PNRs dynamics results in a transition to polar glass state at the so-called freezing temperature, Tf.

We added on page 4 the following text “The strong frequency dispersion of the dielectric permittivity is characteristic for relaxors. It is a manifestation of the slowing down of PNRs dynamics upon cooling, which results in a transition to a short-range ordered polar glass state at the so-called freezing temperature, Tf

  1. Which is the glass transition temperature of the examined systems, and what is the influence of filler on it?

In the case of relaxor one speaks about the polar (dipolar) glass state below the freezing temperature, Tf, which we discussed in our paper.

No other glass transition we have observed in the studied materials, probably due to the limited temperature interval.

  1. How temperature affects the storage efficiency?

The effect of temperature of the storage efficiency is an interesting topic. We have not addressed it in the present study. We expect, that the storage efficiency will increase upon heating, because the polarization hysteresis area will be smaller. At the same time, both polarization and dielectric permittivity drop on heating, correspondingly, the stored and recoverable energy density will decrease.

  1. Since Eq. (5) and (6) fail by far to describe the obtained data (Fig. 3), which is the reason of including them in the paper.

We demonstrated that the increase of dielectric permittivity in composites is stronger as it is predicted by simple mixture models. To comparison we took two the most popular mixture models, Lichtenecker model and Maxwell-Wagner model. To explain the difference, we suggested that the interface effects enhance the dielectric response in nanoparticles-polymer composites.

Reviewer 2 Report

This manuscript reported some test results of dielectric and energy storage performances showing the composites of relaxor terpolymer P(VDF-TrFE-CFE) and BZT nanoparticles. A major revision is needed before acceptance for publishing.

  1. The film preparation process looks like drop coating and flow casting, and these processes are usually not controlled. How did the authors ensure that all the films were around 20 μm thick?
  2. An interesting phenomenon is observed in Figure 2e. Why is that the dielectric loss of the composite films filled with BZT is lower than that of the pure polymer, which increases the permittivity and decreases the dielectric loss?
  3. In addition, in Figure 2a and 2d, the permittivity of two polymers is significantly different, what causes P(VDF-TrFE-CFE) 64.8/35.2/7.8 to have a higher permittivity?
  4. In Figure 5, i don't quite understand why the breakdown strength improved after the introduction of high permittivity BZT into the P(VDF-TrFE-CFE)? What is the mechanism behind the increase in breakdown strength?
  5. The energy storage density is usually regulated by the permittivity and breakdown strength. The permittivity of the composite with 5 vol% BZT is the highest, why is the energy storage density lower than other composites under the same applied electric field?
  6. Please provide some characterization test results to support or assist in illustrating the synthesis and performance improvement of the composites.
  7. Missing experimental details. For example, the synthesis process of BZT, the size of filler, etc.

Author Response

  1. The film preparation process looks like drop coating and flow casting, and these processes are usually not controlled. How did the authors ensure that all the films were around 20 μm thick?

The films are produced via drop coating and solution casting process, which is widely used manufacturing method for various polymers on a lab as well as on industrial scale. The optimized volume of polymer solution is drop coated on a pre-treated glass substrate, having the dimensions of 5x7 cm2. The pre-treatment of glass substrate (by the manufacturer) and the small dimensions of the substrate enable good process control and uniform thickness. The film thickness was controlled using SEM. The relative error of the thickness determination is not more than 5 %.

  1. An interesting phenomenon is observed in Figure 2e. Why is that the dielectric loss of the composite films filled with BZT is lower than that of the pure polymer, which increases the permittivity and decreases the dielectric loss?

Indeed, the dielectric loss of neat polymer below 285 K is slightly higher than of the nanocomposites. Such a minor deviation is most probably an artifact related to the experimental conditions, namely switching of a cooling pump.

  1. In addition, in Figure 2a and 2d, the permittivity of two polymers is significantly different, what causes P(VDF-TrFE-CFE) 64.8/35.2/7.8 to have a higher permittivity?

 This is an interesting point. The processing conditions and parameters are the same for both compositions, therefore it is plausible to conclude that the composition plays a vital role. In our previous publication, Ref. 18, we compared the properties of P(VDF-TrFE-CFE) polymers and found that their dielectric permittivity depends on the composition. This can be related to both the different TrFE/VDF ratio and the different concentration of CFE monomer.

 4. In Figure 5, i don't quite understand why the breakdown strength improved after the introduction of high permittivity BZT into the P(VDF-TrFE-CFE)? What is the mechanism behind the increase in breakdown strength?

We did not perform proper study of the dielectric breakdown strength in the composites and polymers. Therefore, we can not conclude that the breakdown strength is indeed increased in the composites. It is well known that the dielectric breakdown is probabilistic. Therefore in a given experiment, a sample can sustain larger field than its breakdown strength. This was the case for the composites with 1.25 and 5 vol.% of BZT.

  1. The energy storage density is usually regulated by the permittivity and breakdown strength. The permittivity of the composite with 5 vol% BZT is the highest, why is the energy storage density lower than other composites under the same applied electric field?

Indeed, the stored energy density is proportional to the dielectric permittivity. However, we consider the recoverable energy density. For ferroelectric materials it also depends on the value of remanent polarization. The composite with 5% of BZT has larger remanent polarization than the composite with less BZT content. Correspondingly, the recoverable energy density is smaller.

  1. Please provide some characterization test results to support or assist in illustrating the synthesis and performance improvement of the composites.

The synthesis and structural and microstructural properties of nanocomposites have been published separately and is duly cited (reference 38).

  1. Missing experimental details. For example, the synthesis process of BZT, the size of filler, etc.

 The synthesis of BZT nanoparticles and their size distribution is described in details in Ref. 38. We added in “Materials and Methods” the following text. “Details of the synthesis of nanoparticles and their characterization are published elsewhere [38]. The mean size of BZT nanoparticles was 150 nm.”

Reviewer 3 Report

This is a nice manuscript presenting the energy storage properties of polymer-inorganic free-standing composite films based on P(VDF- TrFE-CFE) terpolymer and BaZr0.2Ti0.8O3 nanoparticles. The paper is very interesting for the practical applications. I am fascinated with the clear presentation especially on the Discussion chapter. The paper should be published in the Polymers, however, after some corrections that could increase the quality of the paper:

  1. The conductivity is an important aspect that should be taken into consideration for practical use, such as energy storage applications. Therefore, I consider that the frequency-dependent conductivity spectra need to be included. Alternatively, the impedance plots -Z″ vs. Z′ for each sample may be presented. Some of relevant articles may be considered, such as: ACS Applied Polymer Materials 3, 4869−4878 (2021) and Cellulose 28:843-854 (2021). Please comment on this important point and include the experimental results in the revision.
  2. The authors did not study the mechanical properties of the free-standing composite films. The effect of BZT nanoparticles may affect the mechanical properties and, consequently, the use of the materials.

Author Response

  1. The conductivity is an important aspect that should be taken into consideration for practical use, such as energy storage applications. Therefore, I consider that the frequency-dependent conductivity spectra need to be included. Alternatively, the impedance plots -Z″ vs. Z′ for each sample may be presented. Some of relevant articles may be considered, such as: ACS Applied Polymer Materials 3, 4869−4878 (2021) and Cellulose 28:843-854 (2021). Please comment on this important point and include the experimental results in the revision.

We followed the recommendation of the reviewer and analyzed frequency dependency of conductivity at different temperature. They are shown in Supplementary Information, figures S2-S4. We also added the following text (page 6).

Figures S2 and S3 demonstrate the frequency dependences of ac electrical conductivity measured at different temperatures. Both composite systems exhibit the similar behavior. At temperatures between 320 and 370 K, the ac conductivity increases linearly with frequency, and the temperature variation of the conductivity is relatively weak. Below 310 K, a deviation from the linear dependence becomes visible and the conductivity drops noticeable. Similar behavior has been reported for other polymers [42]. We can attribute this to the slowing down of the PNRs relaxation, which occurs in the same temperature range.

We observed that in the composite films, the conductivity increases with the increasing amount of filler (Figure S4). This correlates with the trend reported in the literature [43, 44]. In addition, the P(VDFx-TrFE1-x-CFEy) 64.8/35.2/7.8 terpolymer has a larger ac conductivity than the P(VDFx-TrFE1-x-CFEy) 68/32/8.5 terpolymer.

  1. The authors did not study the mechanical properties of the free-standing composite films. The effect of BZT nanoparticles may affect the mechanical properties and, consequently, the use of the materials.

Indeed, the mechanical properties of pure polymer and composite films is very interesting topic. This may be the subject of a future study.